# The substrate specificity of the human TRAPPII complex's Rab-guanine nucleotide exchange factor activity

Meredith L. Jenkins[1], Noah J. Harris[1,5], Udit Dalwadi[2,5], Kaelin D. Fleming[1], Daniel S. Ziemianowicz[3], Atefeh Rafiei[4], Emily M. Martin[1], David C. Schriemer [3,4], Calvin K. Yip [2] & John E. Burke [1,2 ✉]

The TRAnsport Protein Particle (TRAPP) complexes act as Guanine nucleotide exchange factors (GEFs) for Rab GTPases, which are master regulators of membrane trafficking in eukaryotic cells. In metazoans, there are two large multi-protein TRAPP complexes: TRAPPII and TRAPPIII, with the TRAPPII complex able to activate both Rab1 and Rab11. Here we present detailed biochemical characterisation of Rab-GEF specificity of the human TRAPPII complex, and molecular insight into Rab binding. GEF assays of the TRAPPII complex against a panel of 20 different Rab GTPases revealed GEF activity on Rab43 and Rab19. Electron microscopy and chemical cross-linking revealed the architecture of mammalian TRAPPII. Hydrogen deuterium exchange MS showed that Rab1, Rab11 and Rab43 share a conserved binding interface. Clinical mutations in Rab11, and phosphomimics of Rab43, showed decreased TRAPPII GEF mediated exchange. Finally, we designed a Rab11 mutation that maintained TRAPPII-mediated GEF activity while decreasing activity of the Rab11-GEF SH3BP5, providing a tool to dissect Rab11 signalling. Overall, our results provide insight into the GTPase specificity of TRAPPII, and how clinical mutations disrupt this regulation.

---

[1] Department of Biochemistry and Microbiology, University of Victoria, Victoria, BC V8W 2Y2, Canada. [2] Department of Biochemistry and Molecular Biology, University of British Columbia, Vancouver, BC V6T 1Z3, Canada. [3] Department of Biochemistry and Molecular Biology, University of Calgary, Calgary, AB T2N 4N1, Canada. [4] Department of Chemistry, University of Calgary, Calgary, AB T2N 4N1, Canada. [5] These authors contributed equally: Noah J. Harris, Udit Dalwadi. ✉email: jeburke@uvic.ca

The ability to transport cargo to specific intracellular membranes is critical to the survival of eukaryotic cells. This process of membrane trafficking is mediated by Rab GTPases, which act as molecular switches, cycling between GTP bound active and GDP bound inactive states. Essential to the regulation of Rab GTPases are the actions of guanine nucleotide exchange factors (GEFs) and GTPase activating proteins (GAPs), which mediate Rab activation and inactivation, respectively[1–3]. Some of the most well-studied Rab GEFs are the large multi-subunit TRAnsport Protein Particle (TRAPP) complexes[4–9]. TRAPP subunits were originally identified in yeast as putative tethering factors mediating vesicle fusion[10–12], and were subsequently found to have GEF activity against the Rab1 and Rab11 homologs in yeast[13,14]. In yeast, three proposed TRAPP complexes exist named TRAPPI (also frequently called the TRAPP core), TRAPPII, and TRAPPIII. TRAPPII is able to act as a GEF for both the yeast Rab1 homolog and the yeast Rab11 homolog[15–17], while TRAPPI and TRAPPIII only have GEF activity for the yeast Rab1 homolog[18–20]. The TRAPPII complex has well-established roles in secretion from the Golgi[17], and the TRAPPIII complex plays key roles in ER-Golgi transport and autophagy[19,21].

Metazoans have two well-established TRAPP complexes, TRAPPII and TRAPPIII, which both share seven conserved subunits (TRAPPC1, TRAPPC2, TRAPPC2L, TRAPPC3, TRAPPC4, TRAPPC5, and TRAPPC6A/B), with two additional subunits for the TRAPPII complex (TRAPPC9 and TRAPPC10) and four additional subunits for the TRAPPIII complex (TRAPPC8, TRAPPC11, TRAPPC12, and TRAPPC13)[22–24]. In metazoans, the specificity of GEF activity for the different TRAPP complexes is conserved in vitro, with TRAPPII acting on both Rab1 and Rab11, and TRAPPIII only having activity on Rab1[24]. The importance of TRAPPII as a Rab11 GEF in vivo in metazoans is highlighted by the interaction of TRAPPII specific subunits with Rab11 in cells[25], and the conditional essentiality of TRAPPII upon knockout of the homolog of the Rab11 GEF SH3BP5[24,26]. In addition, immunoprecipitated TRAPPII has GEF activity for Rab18, with knockout of TRAPPII leading to decreased Rab18 recruitment to lipid droplets[27]. No systematic analysis of GEF activity has been carried out for the mammalian TRAPP complexes against a large panel of Rab GTPases, which suggests the possibility of unknown roles of the TRAPP complexes.

Highlighting the critical roles of the TRAPP complexes in myriad cellular processes has been the discovery of a spectrum of human diseases caused by mutations in different TRAPP subunits, collectively known as TRAPPopathies[5,7,28–35]. Disorders associated with these mutations include the developmental disorder spondyloepiphyseal dysplasia tarda (SEDT), neurodevelopmental delay, microcephaly, epilepsy, and severe intellectual disability. These disorders are caused by mutations, truncations, and deletions in the genes encoding TRAPPC2, TRAPPC2L, TRAPPC6A, TRAPPC6B, TRAPPC9, TRAPPC11, and TRAPPC12, revealing that mutations of both the TRAPPII and TRAPPIII complexes are involved in disease. Intriguingly, many TRAPPopathies share clinical features that occur in loss of function mutations in Rab11[36].

Extensive biochemical, biophysical, and structural studies have provided molecular insight into the mechanism by which the different TRAPP complexes mediate nucleotide exchange. X-ray crystallography and electron microscopy show that the yeast TRAPPI core complex is an extended flat assembly composed of 1 copy of each TRAPPC1, TRAPPC2, TRAPPC4, TRAPPC5, and TRAPPC6, and 2 copies of TRAPPC3[37]. The structure of the yeast homolog of Rab1 bound to the yeast TRAPPI core shows a Rab1 binding interface composed of TRAPPC1, TRAPPC3, and TRAPPC4[38]. Intriguingly, the human homologs appear to be

unable to form a stable TRAPP core complex[37]. The 9 TRAPP subunits of the yeast TRAPPII complex together dimerize to form a compact diamond shape[39], with the dimer stabilized by the Trs65 subunit[40,41]. Trs65 is not conserved in metazoans, and the oligomeric state of the metazoan TRAPPII complex is unknown. The yeast TRAPPIII complex shows a long extended conformation similar to TRAPPI[41,42], which is distinct from TRAPPII. A model of the mammalian TRAPP conserved subunits (TRAPPC1, TRAPPC2, TRAPPC3, TRAPPC4, TRAPPC5, and TRAPPC6) is shown in Fig. 1. The specific molecular determinants for why the mammalian TRAPP complexes achieve differential Rab-GEF specificity are not fully resolved.

The most detailed biochemical and cellular analysis of Rab-GEF specificity has so far only been carried out for the yeast TRAPP complexes. Both TRAPPII and TRAPPIII are more active as Rab GEFs on membrane surfaces[16,20], with activation of TRAPPII driven by Arf1 mediated recruitment to membranes[16,43]. Rab-GEF specificity in yeast is controlled by a steric gating mechanism[15], where the different lengths of the C-terminal hypervariable tails (HVTs) of the different Rab GTPases act to prevent access of Rab1 to TRAPPII, and Rab11 to TRAPPIII on membranes. The length of the HVTs in the mammalian homologs of Rab1 and Rab11 differ slightly than in yeast, and therefore the potential implications of this for Rab-GEF activity are not fully understood. Another proposed mechanism for how TRAPPII can achieve specificity for Rab11 is the presence of a Longin domain in TRAPPC10, which may act as a GEF domain, as the Longin domain of Mon1-Czz1 mediates GEF activity toward Rab7[44,45].

So far there have been no detailed investigations on the Rab GTPase specificity of the mammalian TRAPPII complex, or on the mechanisms through which specificity is achieved. To address this, we have recombinantly expressed and purified in high yield the human TRAPPII complex (TRAPPC1, TRAPPC2, TRAPPC2L, TRAPPC3, TRAPPC4, TRAPPC5, TRAPPC6A, TRAPPC9, and TRAPPC10). Detailed biochemical analysis of 20 different Rab GTPases sampling several Rab families identified GEF activity for Rab1 and Rab11, as well as GEF activity against Rab43 and Rab19, but no activity toward other evolutionarily similar Rab GTPases. Using a combination of electron microscopy, chemical cross-linking, and hydrogen deuterium exchange mass spectrometry (HDX-MS) we have elucidated the architecture of the human TRAPPII complex and show that all Rab GTPases bind at a shared interface. Analysis of clinical mutations in Rab11 show greatly reduced GEF activity, revealing insight into the molecular mechanisms of Rab GTPase mutations in disease.

## Results

**Biochemical analysis of TRAPPII GEF activity**. To investigate the biochemical GEF activity of the mammalian TRAPPII complex we established a method to purify the complex recombinantly in high yield and purity. Complexes were generated using the biGBac multi-promoter expression system in Sf9 insect cells[46], similar to the approach used for the purification of the Drosophila TRAPPII and TRAPPIII complexes[24]. Protein purification was carried out with NiNTA and StrepII affinity columns followed by gel filtration (Fig. 1d, e). Tandem mass spectrometry (MS/MS) analysis of the purified complex identified peptides spanning all expressed subunits. The TRAPPII complex eluted as a highly pure monodisperse species off gel filtration, consistent with the mammalian complex forming the assembly shown in Fig. 1c, which is distinct from the yeast TRAPPII complex, which elutes as a dimer[39].

Guanine nucleotide exchange assays were carried out using Rab isoforms loaded with the fluorescent GDP analog

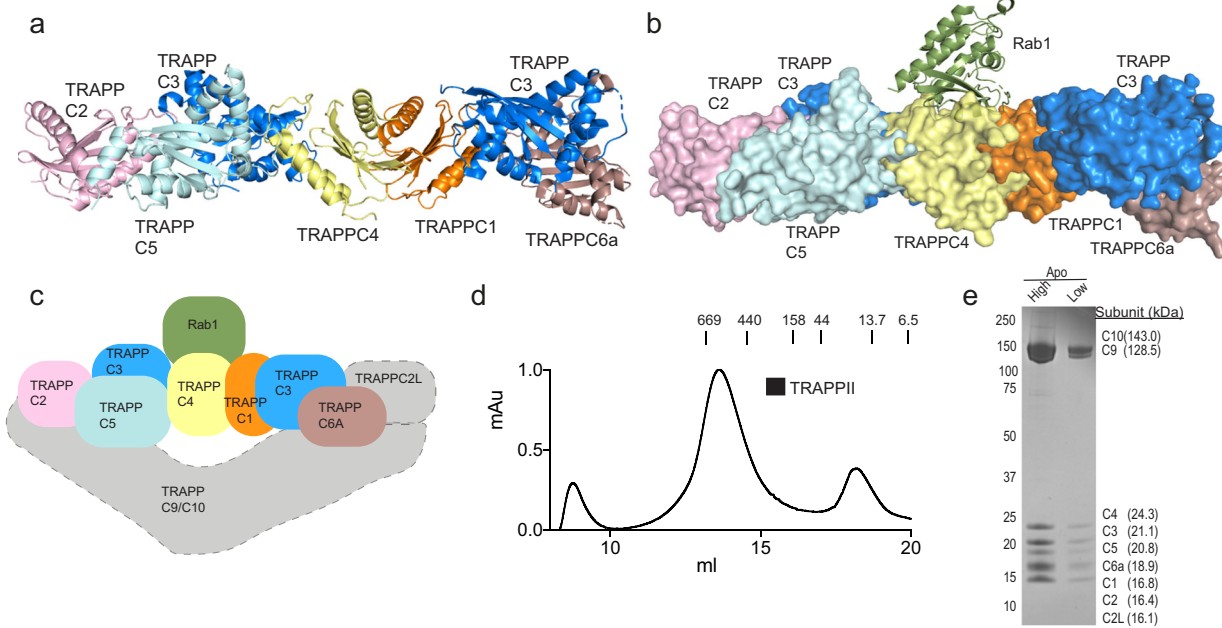

**Fig. 1 Structure models for TRAPP core with and without Rab1 and purification of TRAPPII complex. a** Model of the mammalian TRAPP conserved subunits. The structural model was generated through a combination of the following structures (PDB: 2J3T, 2J3W, and 3CUE)[37,38,78]. Phyre2 was utilized to generate structures of mammalian homologs for TRAPP subunits with no solved crystal structure[79]. The location of TRAPPC2L bound to the conserved subunits is unknown. **b** Surface representation of a model of the human TRAPP core subunits bound to Rab1(PDB:3CUE)[38]. **c** Cartoon model for the human TRAPPII complex. The core subunits are colored according to the models in (**a**) and (**b**). The gray boxes represent TRAPPII specific subunits with an unknown orientation. **d** Size exclusion chromatography (SEC) trace of TRAPPII on a Superose 6 gel filtration column with molecular weight markers indicated (kDa). The y-axis is normalized to max mAU. **e** SDS-PAGE gel of purified TRAPPII complex from the gel filtration peak (~13.5 ml) shown in panel (**d**). High and low labels refer to protein amount loaded (high = 4 μg, low = 0.67 μg).

3-(N-methyl-anthraniloyl)-2-deoxy-GDP (Mant-GDP) and nucleotide exchange was determined as a function of TRAPPII concentration. All Rab GTPases were generated with a C-terminal his tag which allows for the localization on NiNTA-containing synthetic membranes. Initial GEF assays were carried out in solution, with any Rab GTPases showing TRAPPII-mediated GEF activity followed up with experiments on membranes mimicking the Golgi organelle.

The TRAPPII complex had GEF activity for both Rab11 and Rab1 (Fig. 2a, b), with roughly equal catalytic efficiencies (~2.9 × $10^3$ and ~2.0 × $10^3$ M$^{-1}$ s$^{-1}$) for Rab11 and Rab1 (Fig. 2d), respectively. Immunoprecipitated TRAPPII has been shown to have activity on Rab18[27], so we tested the GEF activity of TRAPPII against a panel of Rab GTPases which fully sample the evolutionary history of Rab GTPases. A major focus was on testing the GEF activity of Rab GTPases that are the most evolutionarily similar to Rab1 and Rab11. A total of 18 additional Rab GTPases were tested (Fig. 2b–g). Intriguingly, we found TRAPPII-mediated GEF activity toward Rab43 and Rab19 (Fig. 2b). TRAPPII showed increased catalytic efficiency for Rab43 compared to Rab1 or Rab11 (~11.1 × $10^3$ M$^{-1}$ s$^{-1}$ vs ~2.9 and ~2.0 × $10^3$ M$^{-1}$ s$^{-1}$, respectively) (Fig. 2d). Attempts to measure the catalytic efficiency for Rab19 were unsuccessful, as there was a non-linear response of GEF activity with increasing TRAPPII concentration (Fig. 2b). Due to this complication, and that Rab43 and Rab19 are very evolutionarily similar to each other, only diverging from each other in vertebrata[47], only Rab43 was analyzed further. Rab43 is an essential Golgi localized GTPase[48] that plays important roles in GPCR trafficking[49], with no currently identified GEF. Rab19 is a paralog of Rab43, which is also localized to the Golgi[50,51]. There was no detectable TRAPPII-mediated GEF activity for any other Rab GTPase, including Rab18 or the Rab11 isoform Rab25 (also known as Rab11C)

(Fig. 2c). As Rab18 was previously annotated as a TRAPPII substrate, we tested GEF activity in the presence and absence of membranes, which showed no detectable GEF activity (Fig. 2h). The strict selectivity of TRAPPII for Rab11A over Rab25 is intriguing, as the only other known human Rab11 GEF SH3BP5 has roughly equal GEF activity for both[52].

Both the *Drosophila* and yeast variants of TRAPPII were more efficient Rab GEFs when the Rab GTPase was presented on a membrane surface[16,24]. We tested Rab1, Rab11, and Rab43 on artificial membranes mimicking the Golgi and found increased GEF activity for both Rab1 and Rab11, but no difference in GEF activity against Rab43 (Fig. 2e, f). We tested a panel of different vesicles for any difference in Rab11 GEF activity and found that membranes containing PC did not alter GEF activity, but the GEF activity was increased as the surface charge was increased (Fig. 2e). This is consistent with the role of anionic lipids in increasing TRAPPII activity[16].

**Structural studies of the TRAPPII complex and its binding Rab GTPases.** To further verify that TRAPPII can act as a GEF toward Rab43, we carried out gel filtration experiments with the TRAPPII complex in the presence and absence of GST-tagged Rab43. The tagged GTPases co-eluted with the TRAPPII complex (Fig. 3a, b). The Rab1 and Rab11 GEF activity of the yeast TRAPPII complex is mediated by the canonical Rab-binding site composed of the TRAPPC1 and TRAPPC4 subunits[16]. We postulated that the activity toward Rab43 may be mediated by the putative Longin domain present in TRAPPC10, as proposed for Rab11[53]. We used hydrogen deuterium exchange mass spectrometry (HDX-MS) to define the interface of TRAPPII with all three Rab GTPases (Rab1, Rab11, Rab43). These experiments were performed in the presence of EDTA to generate a

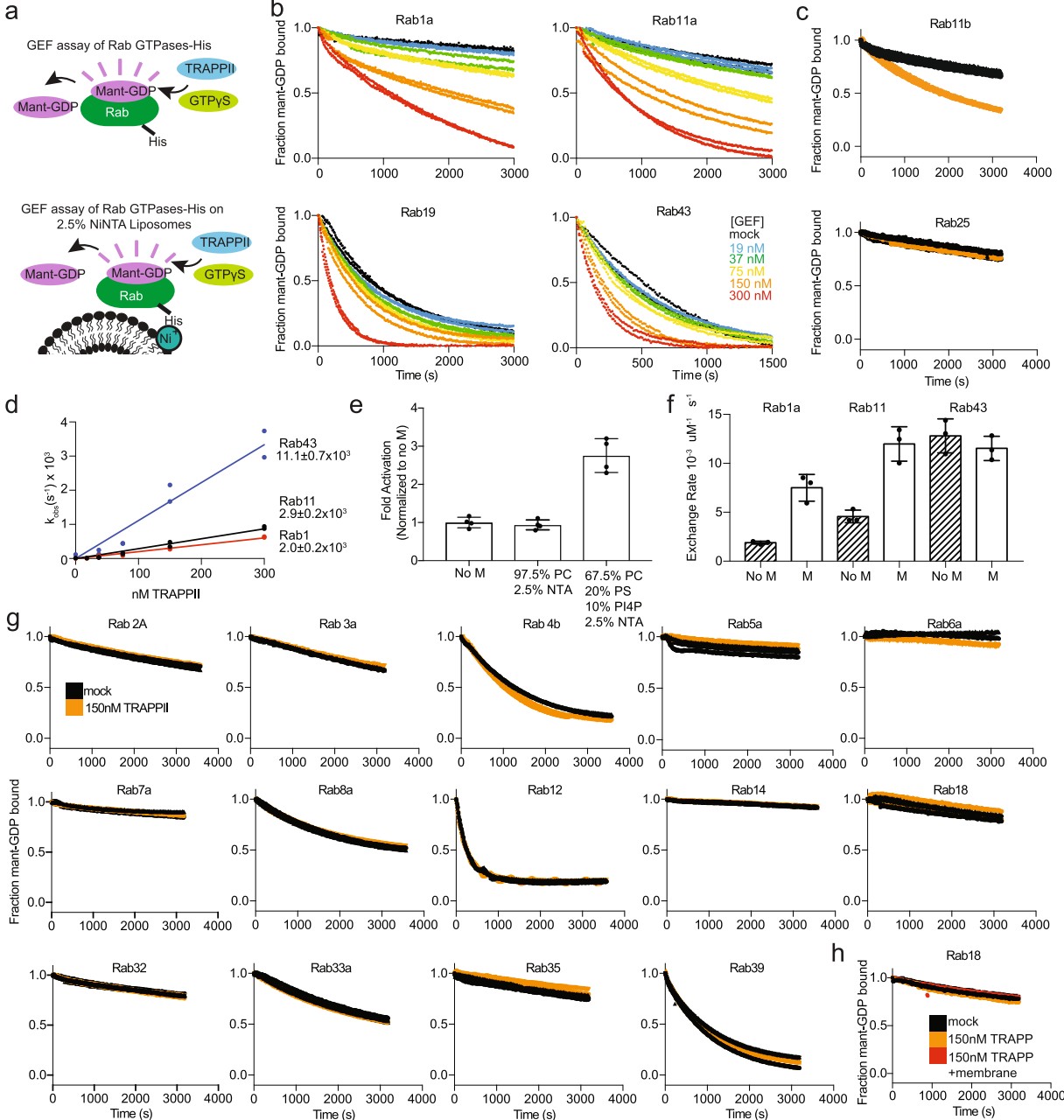

**Fig. 2 In vitro GEF assays reveal that TRAPPII is a potent GEF for Rab1, Rab11, Rab19, and Rab43. a** Cartoon Schematic of GEF activation assays using fluorescent analog Mant-GDP in the presence and absence of NiNTA-containing lipid vesicles. **b** In vitro GEF assay of TRAPPII on Rab1a, Rab11a, Rab19, and Rab43. Nucleotide exchange was monitored by measuring the fluorescent signal during the TRAPPII (19–300 nM) catalyzed release of Mant-GDP from 4 μM of Rab-His6 in the presence of 100 μM GTPγS. Each concentration was conducted in duplicate ($n = 2$). **c** In vitro GEF assay of TRAPPII on Rab11b and Rab25(Rab11c). Nucleotide exchange was monitored by measuring the fluorescent signal during the TRAPPII (150 nM) catalyzed release of Mant-GDP from 4 μM of Rab-His6 in the presence of 100 μM GTPγS. Error bars represent SD ($n = 3$). **d** Nucleotide exchange rates of Rab1, Rab11, and Rab43 plotted as a function of TRAPPII concentration. The kcat/Km values for all Rabs were calculated from the slope ($n = 2$). **e** Bar graph representing the difference in Rab11 GEF activity in the presence and absence of two different 400 nm extruded liposomes at 0.2 mg/ml. Nucleotide exchange was monitored by measuring the fluorescent signal during the TRAPPII (150 nM)-catalyzed release of Mant-GDP from 4 μM of Rab11-His6 in the presence of 100 μM GTPγS. Error bars show SD ($n = 4$). **f** Bar graph representing the difference in GEF activation of Rab1, Rab11, and Rab43 in the presence and absence of 400 nm extruded liposomes at 0.2 mg/ml (67.5% PC, 20% PS, 10% PI(4)P, 2.5% DGS NTA). Error bars show SD ($n = 3$). **g** In vitro GEF assays of TRAPPII against a panel of 14 Rab GTPases loaded with Mant-GDP with 150 nM TRAPPII and 4 μM Rab GTPase ($n = 2$–3). **h** In vitro GEF assay of 4 μM Rab18 loaded with Mant-GDP with 150 nM TRAPPII in the presence or absence of 400 nm extruded liposomes at 0.2 mg/ml (67.5% PC, 20% PS, 10% PI(4)P, 2.5% DGS NTA) ($n = 3$).

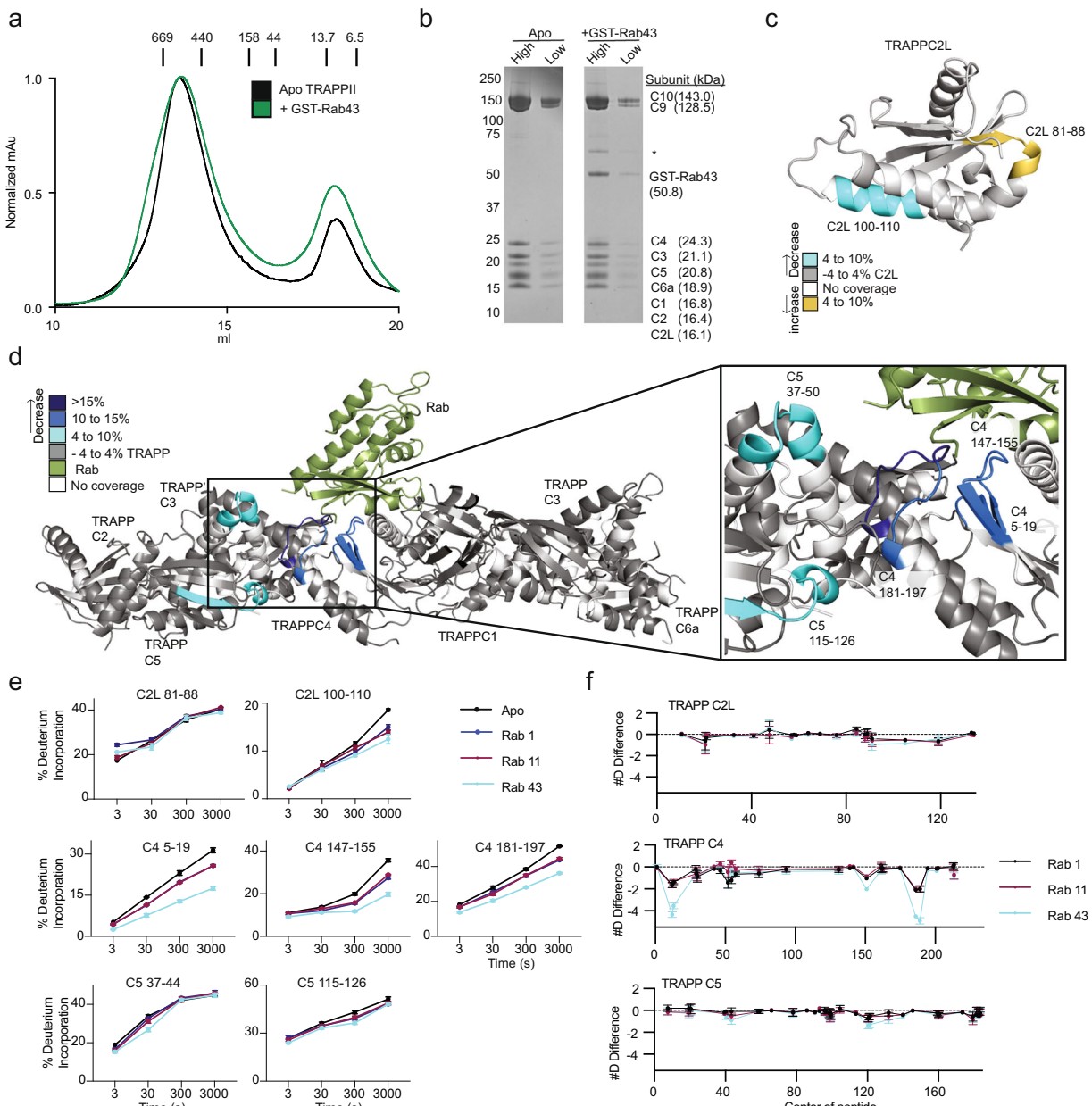

**Fig. 3 HDX-MS reveals a conserved Rab-binding interface for TRAPPII. a** Size Exclusion Chromatography (SEC) trace of the TRAPPII and TRAPPII: GST-Rab43 complexes. Apo proteins, and proteins mixed at a 2:1 molar ratio, were subject to SEC on a Superose 6 Increase 10/300 GL column. Size markers are indicated on the trace (kDa). Rab43 and TRAPPII co-eluted together in a single peak, indicating complex formation. **b** An SDS-PAGE gel of each SEC peak is shown (4–12% NuPAGE gradient gel run at 200 V for 45 min and stained with Coomassie Brilliant Blue dye). High and low labels refer to protein amount loaded (high = 4 μg, low = 0.67 μg). The asterisk denotes a contaminating *E.coli* protein. **c** Maximum significant differences in HDX observed across all time points upon Rab43 incubation with TRAPPII are mapped on the predicted structure of TRAPPC2L. Peptides are colored according to the legend. **d** Maximum significant differences in HDX observed across all time points upon Rab43 incubation with TRAPPII are mapped on the model of the TRAPP core. Peptides are colored according to the legend. **e** Selected TRAPPII peptides displaying decreases in exchange in the presence of either Rab1, Rab11, or Rab43 are shown. For all panels, error bars show SD (*n* = 3). **f** The number of deuteron difference for all analyzed peptides over the entire deuterium exchange time course for Rab1, Rab11, or Rab43 in the presence of TRAPPII. Only subunits with significant differences are shown. Every point represents the central residue of an individual peptide. Changes are mapped according to the legend.

nucleotide-free stabilized Rab-GEF complex. HDX-MS is a powerful analytical technique that measures the exchange rate of amide hydrogens with deuterated buffer, and since the exchange rate of amide hydrogens is primarily governed by secondary structure dynamics[54], it is therefore an excellent readout of protein dynamics[55,56].

HDX-MS experiments were carried out with TRAPPII in the presence and absence of a 3-fold excess Rab1, Rab11, and Rab43.

Deuterium incorporation in HDX-MS is measured by the generation of pepsin peptide fragments, with coverage for all TRAPPII subunits obtained (TRAPPC1, TRAPPC2, TRAPPC2L, TRAPPC3, TRAPPC4, TRAPPC5, TRAPPC6A, TRAPPC9, and TRAPPC10). Significant decreases in deuterium incorporation (defined as >4%, >0.4 Da, two-tailed T-test *p* value < 0.01) were identified in the TRAPPC2L, TRAPPC4, and TRAPPC5 in the presence of Rab GTPases with no other changes in other TRAPP

subunits (Fig. 3c–f, Supplemental Fig. 2). The majority of these changes mapped onto the canonical Rab interface[38]. The largest decreases in exchange occurred in the TRAPPC4 subunit, in a peptide spanning 181–191, which would likely be in contact with both the N and C termini of the Rab substrate. Regions in TRAPPC5 with decreased exchange are not proposed to be in direct contact with Rab substrates, however, these regions are in contact with the c-terminal region of TRAPPC4, likely highlighting an allosteric conformational change that accompanies Rab binding. The TRAPPC1 subunit, which was found to interact with Rab in the yeast TRAPP core structure, showed no decreases in exchange, however, this helix is extremely stable, with less than 10% exchange even at 3000 s, which may explain why no significant difference is observed with Rab substrate. The TRAPPC2L subunit exhibited both decreases and increases in exchange that occurred (81–88, 100–110, respectively). As this subunit is proposed to be quite distant from the Rab-binding site, this is likely due to an allosteric effect, with an unknown molecular mechanism. Comparing the three Rab substrates tested (Rab1, Rab11, and Rab43), the largest decrease in exchange was observed with Rab43, potentially indicating a higher affinity.

To further define the architecture of the TRAPPII complex and the structural basis for Rab-GEF activity we subjected recombinant human TRAPPII to negative-stain single-particle electron microscopy (EM) analysis. Initial images revealed triangular-shaped particles (Fig. 4a, b) distinct from the previously characterized diamond-shaped yeast TRAPPII[39], suggesting a strictly "monomeric" assembly as postulated. Two-dimensional (2D) analysis and 3D reconstruction confirmed this observation and further showed that the triangular-shaped human TRAPPII contains an elongated and well-defined peripheral region reminiscent of yeast TRAPPI, which we referred to as TRAPP core (TRAPPC1, TRAPPC2, TRAPPC3, TRAPPC4, TRAPPC5, and TRAPPC6A). This peripheral region is capped by two prominent "flaps" that appear to extend toward and intersect at a less well-defined and likely conformationally flexible base (Fig. 4c). To delineate the Rab43 binding site on human TRAPPII, we examined the purified TRAPPII-Rab43 complex by negative-stain EM. Two-dimensional analysis showed that the additional density contributed by GST-tagged Rab43 is located at the TRAPP core (Supplemental Fig. 3), which is consistent with results obtained from HDX-MS analysis (Fig. 4d, e).

We utilized chemical cross-linking mass spectrometry to define the architecture of the different TRAPP subunits relative to each other. We implemented the hetero-bifunctional photo-activatable cross-linkers LC-SDA and SDA, as these provide advantages in minimizing kinetic trapping of spurious non-specific interactions[57]. Initial analysis of a model of the TRAPP core revealed 27 cross-links, of which 26 were consistent with the structural model of the TRAPP core (Supplemental Data 2). These included cross-links between TRAPPC1/TRAPPC4, TRAPPC1/TRAPPC3, TRAPPC3/TRAPPC5, TRAPPC3/TRAPPC6A, and TRAPPC5/TRAPPC2 (Fig. 4f, g). The TRAPPC2L subunit only had cross-links with a single core subunit (TRAPPC6A), consistent with previous data suggesting an interaction of TRAPPC2L with TRAPPC3-TRAPPC6[58]. Multiple cross-links were observed for both TRAPPC9 and TRAPPC10 with multiple TRAPP subunits, including a number of direct TRAPPC9 and TRAPPC10 cross-links. The TRAPPC10 subunit had interactions with the TRAPPC6 and TRAPPC2L subunits on the one side of the TRAPP core. Surprisingly, the TRAPPC2 and TRAPPC5 subunits located on the other side of the TRAPP core had multiple interactions with both TRAPPC9 and TRAPPC10, suggesting that these proteins likely form an extended shared interface.

**Analysis of clinical and engineered mutations in Rab11 and Rab43.** Clinical mutations in Rab11 share some similar phenotypes with TRAPPopathies, with mutations in Rab11A and Rab11B leading to developmental disorders[36,59]. GEF activity assays on the K13N mutation in Rab11 showed that this mutation completely disrupted TRAPPII-mediated activity, similar to what was observed for the Rab11 GEF SH3BP5[52], which reveals that this mutation completely abrogates the ability of Rab11 to be activated by any identified Rab11 GEF (Fig. 5a, b).

The ability for TRAPPII to activate Rab43 led us to investigate the potential consequences of Rab43 phosphorylation in modulating TRAPPII GEF activity. The Parkinson's disease-linked kinase LRRK2 can phosphorylate different Rab GTPases on a conserved threonine in switch II, with Rab43 identified as an endogenous LRRK2 substrate[60]. The phosphorylation of Rab8 in switch II led to a ~4-fold decrease in activation by its cognate GEF Rabin8[61]. To examine if this was conserved in Rab43 we sought to generate phosphorylated and phosphomimetic variants of Rab43. We attempted to generate phosphorylated Rab43 using the kinase MST3, which was used to generate high yields of phosphorylated Rab8 for structural studies with its effector RILPL2[62], however, there was no detectable Rab43 phosphorylation in these samples. We generated a phosphomimetic T82D mutation in Rab43 and found ~2-fold decreased TRAPPII-mediated GEF activity compared to wild-type Rab43 (Fig. 5c–e). This highlights a potentially conserved role of switch II phosphorylation in modulating Rab activation.

There are two established GEFs for Rab11, SH3BP5 and the TRAPPII complex, with either becoming conditionally essential if the other is knocked out in *Drosophila*[24]. However, there are likely non-redundant roles of these GEFs with clinical significance, as disease-linked mutations do occur in TRAPPII unique subunits. Deciphering non-redundant roles of TRAPPII compared to SH3BP5 would be assisted by the generation of Rab11 mutations that selectively blunt activation by only one GEF. From examining the structures of Rab11 bound to SH3BP5[52] and the Rab1 homolog bound to the yeast TRAPP core[38] we identified K58 in Rab11 as a potential residue to mutate to disrupt SH3BP5 mediated activity, while having a limited effect on TRAPPII (Fig. 5f). In the SH3BP5-Rab11 complex there are a number of negatively charged residues in Rab11 and SH3BP5 that are in vicinity of the K58 residue in the inter-switch region in Rab11. There are no negatively charged residues in the vicinity of this area in the TRAPP core.

We postulated that a charge reversal mutation K58E could alter the Rab-GEF rate in SH3BP5 but have only a limited effect on TRAPPII-mediated exchange. Biochemical assays on the K58E variant of Rab11 showed a ~10-fold decrease in SH3BP5 GEF mediated nucleotide exchange, with no significant difference in TRAPPII-mediated activation both in solution and on membranes (Fig. 5g, h). Importantly, the K58 residue is distant from the proposed interface for Rab11 GAPs, escort proteins, and effector binding proteins (Fig. 5i), highlighting the potential of this mutation as a tool to define non-redundant roles of the Rab11 GEFs in specific cells and tissues.

## Discussion

The TRAPP complexes are important regulators of membrane trafficking in eukaryotes. They have been well characterized in yeast, and play major roles in secretion, ER-Golgi transport, and autophagy[17,19,21]. Extensive biochemical and biophysical studies on the yeast TRAPPII and TRAPPIII complexes have revealed key insight into their Rab-GEF selectivity, and the molecular basis for their regulation[15–17,20,38,39,41,42,63]. Although the yeast TRAPP complexes are well characterized, there are several

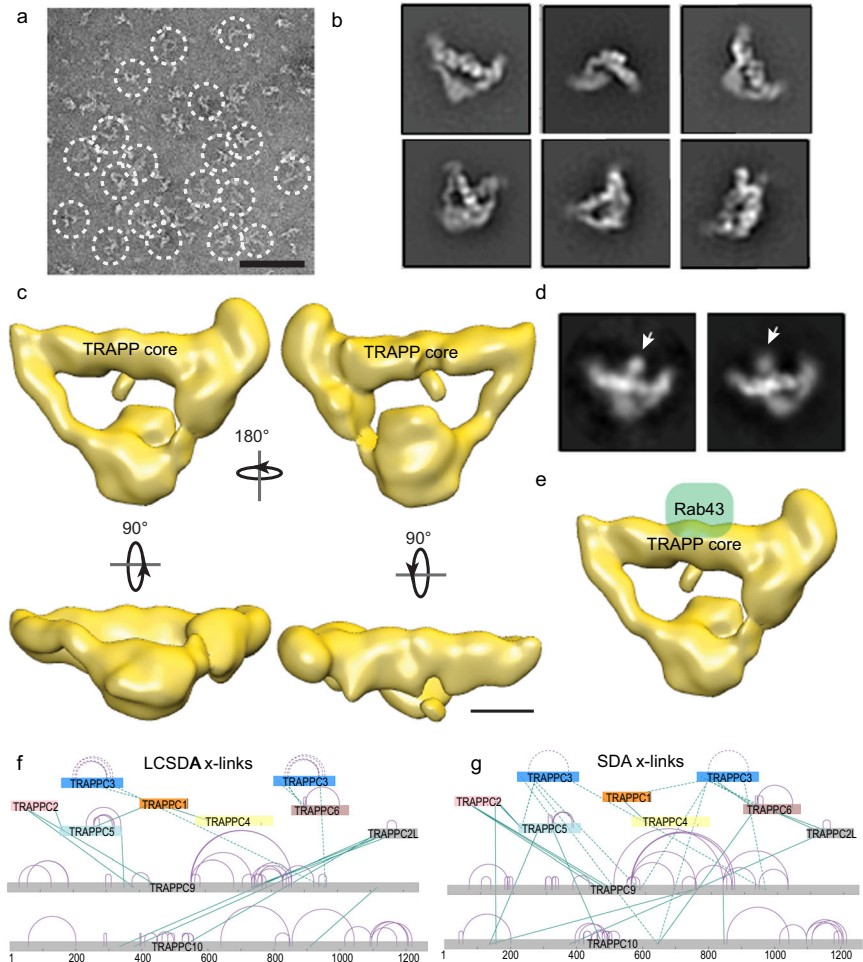

**Fig. 4 Electron microscopy and chemical cross-linking reveal architecture of the TRAPPII complex. a** A representative raw image of negatively stained purified human TRAPPII complex. Triangle-shaped particles but not diamond-shaped particles were observed (circled). Scale bar is 50 nm. **b** Representative 2D class averages of TRAPPII complex. Box edge length of 45 nm. **c** 3D reconstruction of TRAPPII with multiple orientations of the complex. Scale bar is 50 Å. **d** Representative 2D class averages of TRAPPII complex with GST-tagged Rab43. The Rab43 density localizes to the center of the TRAPP core (arrow). Box edge length of 45 nm. **e** EM density maps of TRAPPII with a modeled position of Rab43. Scale bar is 50 Å. **f, g** Reduced set of cross-linking mass spectrometry data for the TRAPPII complex using the hetero-bifunctional photo-activatable cross-linkers NHS-Diazirine (SDA), and NHS-LC-Diazirine (LC-SDA). The TRAPP subunits are arranged similar as in the orientation of Fig. 1a. Inter-protein cross-links are shown in green, with intra-protein cross-links shown in purple. Cross-links to the TRAPPC3 subunit are shown as dashed lines to highlight the ambiguity arising from two copies. Complete cross-linking data are shown in the source data. Cross-link map networks are generated with xiNET online cross-link viewer tool[80].

differences between the composition and identity of the proteins that make up the yeast and mammalian TRAPP complexes, with limited analysis of the mammalian variants. A primary function of the TRAPP complexes is their ability to work as GEFs for Rab GTPases, therefore, defining the Rab substrate selectivity of mammalian TRAPP complexes is essential to understand their cell-specific roles in membrane trafficking. Our biochemical and biophysical analysis of the mammalian TRAPPII complex has revealed insight into its activity on previously uncharacterized Rab substrates, the molecular basis for how it is assembled, and how clinical mutations can alter its regulation.

Previous biochemical analysis of the *Drosophila* TRAPPII complex revealed that it was composed of TRAPPC1, TRAPPC2, TRAPPC3, TRAPPC4, TRAPPC5, TRAPPC6, TRAPPC9, and TRAPPC10, and had biochemical activity against both Rab1 and Rab11[24]. Biochemical analysis of the mammalian variants of TRAPPII using immunopurified material of unknown composition revealed limited activity against Rab11[64], and putative activity toward Rab18[27]. Our study using recombinant homogenous mammalian TRAPPII has allowed for detailed

biochemical analysis of Rab substrate specificity. We find that this complex has potent activity against both Rab1 and Rab11, and furthermore shows activity against Rab19 and Rab43, which are Golgi localized and evolutionarily similar to Rab1[47]. There are no previously identified Rab GEFs for either Rab19 or Rab43, with Rab43 playing a critical role in mediating GPCR trafficking to the plasma membrane[49]. We detected no GEF activity toward Rab18, similar to the results of the *Drosophila* TRAPPII complex[24]. This discrepancy may be explained by additional TRAPP binding partners that complicate analysis of immunopurified material. The protein C7orf43, otherwise known as TRAPPC14, was identified as a binding partner to TRAPPII that could also bind to the Rab8 GEF Rabin8[65]. It is possible that this or an additional co-purified GEF may mediate the observed Rab18 GEF activity. Intriguingly, the TRAPPII complex had no detectable activity toward the Rab11 family member Rab25 (also known as Rab11c), with this specificity being distinct from the previously identified Rab11 GEF SH3BP5[52]. From observing the alignment of Rab25 to TRAPPII-Rab substrates (Rab1, Rab11, Rab19, and Rab43) the most divergent putative TRAPPII contact site is the N-terminus

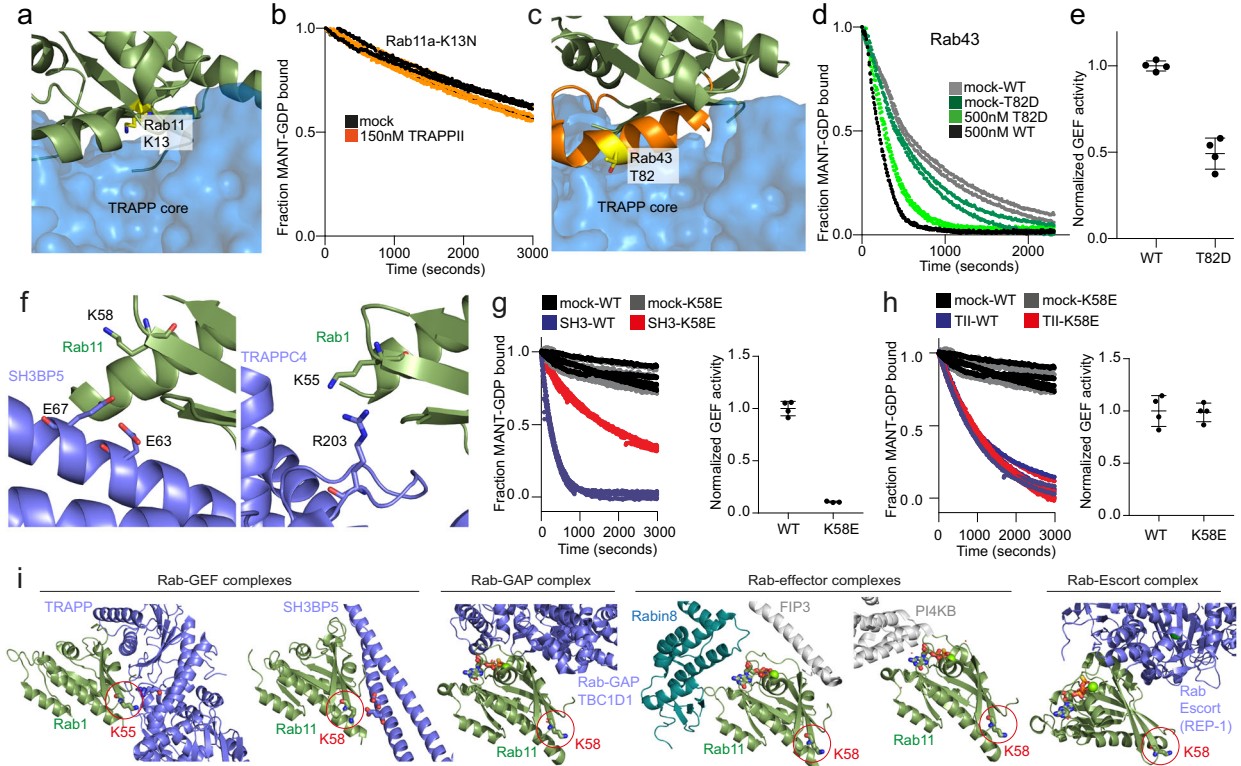

**Fig. 5 Biochemical analysis of clinical Rab mutants shows specific altered GEF activity. a** Model of Rab11 bound to the TRAPP core, with the K13 residue that is mutated in developmental disorders (K13N). **b** In vitro GEF assay of TRAPPII WT (150 nM) for the clinically relevant mutation Rab11a K13N (4 μM) (n = 3). **c** Model of Rab43 bound to the TRAPP core, highlighting the residue T82 which can be phosphorylated by LRRK2. **d** Normalized GEF activity showing TRAPPII activation of WT Rab43 and Rab43-T82D phosphometic mutant (n = 2). **e** The rates of GEF exchange for WT Rab43 and phosphomimetic T82D normalized against WT. Error bars show SD (n = 4). **f** Model of SH3BP5 and the TRAPP core bound to Rab11 and Rab1, respectively (based on PDB models PDB:3CUE, 6DJL)[38,52]. The modified residue (K58) is labeled on both. **g** (left) Normalized GEF activity trace showing SH3BP5 (150 nM) activation of Rab11 WT (4 μM) and Rab11-K58E (4 μM) mutant (n = 3). (right) Rates of exchange were calculated and normalized against WT to determine the effect of the K58 mutant activation by SH3BP5. Error bars show SD (n = 4). **h** (left) Normalized GEF activity showing TRAPPII (150 nM) activation of Rab11 WT (4 μM) and Rab11-K58E (4 μM) mutant (n = 3). (right) Rates of exchange were calculated and normalized against WT to determine the effect of the K58 mutant activation by TRAPPII. Error bars show SD (n = 4). **i** Analysis of the interface of Rab11 with GEFs, GAPs, escort, and effector proteins compared to the location of the mutated K58 residue. The GEF complexes are from the structures of yeast TRAPP core bound to Rab1 (PDB: 3cue), and SH3BP5 bound to Rab11 (PDB: 6DJL)[38,52]. The GAP model was generated by aligning Rab11 with the Rab33 portion of the gyp1/Rab33 complex (PDB: 2G77)[81]. The structure of the TBC domain of the human TBC1D1 (PDB: 3QYE)[82] was then superimposed on the Gyp1p TBC domain, and the cartoon TBC1D1 TBC domain is illustrated (blue). The effector models are from the structure of Rab11 bound to FIP3 and Rabin8, and Rab11 bound to PI4KB (PDB: 4UJ4, 4D0L)[83,84]. The escort complex was generated by superimposing the Rab7 component of a REP-1/Rab7 complex (PDB: 1VG0) onto Rab1[85].

leading into the first beta strand. Further additional high-resolution structural information on Rab binding to TRAPPII will be critical to define the molecular mechanism of specificity. There was increased GEF activity for the mammalian TRAPPII complex when Rab substrates were presented on anionic membranes, but this was to a much lower extent than observed with the yeast TRAPPII complex[16]. The molecular mechanism of increased activity is likely driven by increased membrane recruitment as seen in the yeast TRAPPII complex[16], although further experiments will be required to see if allosteric effects may also play a role.

The observation of Rab-GEF activity of the TRAPPII complex against Rab43 and Rab19 suggested that there may be a Rab-binding site distinct from the canonical TRAPPC4 and TRAPPC1 binding site, potentially mediated by the Longin domain present in TRAPPC10[53]. HDX-MS experiments revealed a conserved Rab-binding site for Rab1, Rab11, and Rab43 that mainly composed of TRAPPC4 and TRAPPC1, consistent with the Rab-binding site identified from the crystal structure of the yeast homolog of Rab1 bound to the yeast TRAPP core[38]. Electron microscopy and chemical cross-linking studies allowed for

detailed analysis of the architecture of the mammalian TRAPPII complex. In particular, our 2D and 3D EM analysis clearly showed that TRAPPII has an overall triangle shape that is consistent with a "monomeric" ~430 kDa TRAPPII complex. The Trs65 subunit which mediates dimerization in yeast TRAPPII is absent in higher-order organisms[41], and this may explain the lack of a high-order oligomeric state observed for the mammalian complex. EM analysis of Rab43 bound to TRAPPII provided additional evidence supporting the conserved Rab-binding site of TRAPPII. Chemical cross-linking data of the TRAPPII complex confirmed expected interactions of the TRAPP core suggested by previous crystallographic analysis[37], and revealed the binding site of TRAPPC2L to TRAPPC6a. Intriguingly, cross-linking also showed interactions between TRAPPC10 with either end of the TRAPP core (TRAPPC2 and TRAPPC2L), which suggests this subunit forms a large extended interface. TRAPPC9 also interacted with TRAPPC2, which suggests an interweaving interaction between TRAPPC9/TRAPPC10. This could explain why biochemical and EM experiments performed on the yeast TRAPPII complex have suggested varying arrangements of TRAPPC10 and TRAPPC9 with respect to the TRAPP core[5,39,40,58,66,67]. The

molecular determinants of how the TRAPP core precisely interacts with the TRAPPII specific subunits will have to await a high-resolution structural analysis of TRAPPII.

The key roles of the TRAPP complexes in human health are underscored by multiple clinical mutations in numerous TRAPP subunits that cause a spectrum of pathological conditions[5,68]. An interesting observation is the similarity between clinical symptoms seen in patients with TRAPP mutations and those with Loss of function mutations in Rab11A/B[36,59]. The K13N mutation in Rab11 leads to an almost complete disruption in TRAPPII-mediated GEF activity, which is similar to the disruption of SH3BP5 mediated GEF activity[52]. This indicates that the K13N completely abrogates the ability of Rab11 to be activated by either TRAPPII or SH3BP5. Furthermore, multiple Rab GTPases can be phosphorylated by the Parkinson's related kinase LRRK2, with this leading to disruption in activation by Rab GEFs and effector binding[60–62]. Rab43 is endogenously phosphorylated by LRRK2 at T82 in switch II, and biochemical assays of the phosphomimetic T82D mutation showed a roughly 2-fold decrease in GEF activity consistent with decreases seen for phosphorylated Rab8 activation by Rabin8[61].

Critical to fully understanding the roles of the TRAPPII complex is defining how it regulates the localization and activation of Rab11. Studies in metazoans have shown that neither TRAPPII specific subunits nor the Rab11 metazoan specific GEF SH3BP5 are essential, but the knockout of both is lethal[24]. However, in humans, the important roles of the TRAPPII complex are highlighted by multiple disease-linked mutations in TRAPPC9[5]. Essential to deciphering the specific roles of different GEFs in mediating Rab11 activation is the generation of Rab mutants that can only be activated by a specific Rab GEF. We identified a negative charge cluster in SH3BP5 that was not present in TRAPPII. We mutated K58 in Rab11, which is in the vicinity of the negatively charged patch in SH3BP5 and found that this led to a ~10-fold decrease in SH3BP5 GEF activity, with no significant decrease in TRAPPII-mediated GEF activity. This mutation will act as a valuable tool to investigate the cell and tissue-specific roles of TRAPPII and SH3BP5.

The TRAPP complexes are key regulators of membrane trafficking. The detailed biochemical and biophysical experiments presented here provide a framework for understanding the activity and regulation of the TRAPP complex. In addition, Rab11 GEF selective mutants provide a mechanism to define roles of TRAPPII in regulating Rab11 specific cellular functions.

## Methods

**Plasmids and antibodies.** The full-length human TRAPP genes, TRAPPC1 (HsCD00337916), TRAPPC2L (HsCD00340414), and TRAPPC10 (HsCD00341380) were purchased from the Dana Farber Plasmid Repository. The full-length human TRAPP genes, TRAPPC2 (HsCD00040385), TRAPPC6a (HsCD00674667), TRAPPC4 (HsCD00396892), TRAPPC5 (HsCD00398807), TRAPPC6b (HsCD00352944), and TRAPPC9 (HsCD00820727) were purchased from the DNASU. The full-length human TRAPPC3 gene (Plasmid #34711) was purchased from AddGene. Genes were subcloned into pLIB vectors for expression with no engineered tags, while in the case of TRAPPC3 a TEV cleavable C-terminal 2x strep tag was added, and a C-terminal 6x his tag was added to TRAPPC10. Genes were subsequently amplified following the biGBac protocol to generate two plasmids, each containing 4–5 TRAPP genes[46]. The following full-length human Rab genes were obtained from AddGene (plasmid #), Rab1 (49467), Rab3a (49542), Rab6a (49469), Rab18 (49550), and Rab33 (49551). Rab2a (HsCD00383517), Rab4b (HsCD00296539), Rab8a (HsCD00044586), Rab12 (HsCD00297182), Rab14 (HsCD00322387), Rab25 (HsCD00327461), Rab35 (HsCD00327461), Rab39 (HsCD00335627), and Rab43 (HsCD00334332) were purchased from the Dana Farber Plasmid Repository. Rab7a (HsCD00829591), Rab19 (HsCD00632710), and Rab32 (HsCD00000686) were purchased from DNASU. Genes were subcloned into pOPTGcH vectors for expression with an n-terminal cleavable GST tag and a non-cleavable c-terminal his tag.

**Protein expression.** All TRAPPII complexes were expressed in Sf9 cells, using an equal amount of the appropriate vectors (Supplemental Table 1). In brief, an optimized ratio of baculovirus was used to co-infect *Spodoptera frugiperda* (Sf9) cells between 1 and $2 \times 10^6$ cells/mL. Co-infections were harvested at 66-h and washed with ice-cold PBS before snap-freezing in liquid nitrogen. Rab constructs were all expressed in BL21 C41 *E. coli*, induced with 0.5 mM IPTG and grown at 37 °C for 4 h. For optimal yield, Rab19 and Rab43 were expressed in BL21 C41 *E. coli* induced with 0.1 mM IPTG and grown overnight at 23 °C. Pellets were washed with ice-cold phosphate-buffered saline (PBS), flash-frozen in liquid nitrogen, and stored at −80 °C until use.

**Protein purification.** TRAPPII cell pellets were lysed by sonication for 1.5 min in lysis buffer (20 mM Tris pH 8.0, 100 mM NaCl, 5% (v/v) glycerol, 2 mM ß-mercaptoethanol (BME), and protease inhibitors (Millipore Protease Inhibitor Cocktail Set III, Animal-Free)). Triton X-100 was added to 0.1% v/v, and the solution was centrifuged for 45 min at 20,000 × g at 1 °C. The supernatant was then loaded onto a 5 mL HisTrap™ FF column (GE Healthcare) that had been equilibrated in NiNTA A buffer (20 mM Tris pH 8.0, 100 mM NaCl, 10 mM imidazole pH 8.0, 5% (v/v) glycerol, 2 mM bME). The column was washed with 20 mL of NiNTA buffer, 20 mL of 6% NiNTA B buffer (20 mM Tris pH 8.0, 100 mM NaCl, 200 mM imidazole pH 8.0, 5% (v/v) glycerol, 2 mM bME) before being eluted with 100% NiNTA B. The eluate was subsequently loaded on a 5 ml Strep™ column and washed with 10 ml SEC buffer (20 mM HEPES pH 7.5, 150 mM NaCl, 5% glycerol, 0.5 mM TCEP). The Strep-tag was cleaved by adding SEC buffer containing 10 mM BME and TEV protease to the column and incubating overnight at 4 °C. Protein was pooled and concentrated using an Amicon 50 K concentrator and size exclusion chromatography (SEC) was performed using either a Superose 6 increase 10/300 column or a Superdex 200 increase 10/300 column equilibrated in SEC buffer. Fractions containing protein of interest were pooled, concentrated, flash-frozen in liquid nitrogen and stored at −80 °C.

For Rab purification, cell pellets were lysed by sonication for 5 min in lysis buffer (20 mM Tris pH 8.0, 100 mM NaCl, 5% (v/v) glycerol, 2 mM ß-mercaptoethanol (BME), and protease inhibitors (Millipore Protease Inhibitor Cocktail Set III, Animal-Free)). Triton X-100 was added to 0.1% v/v, and the solution was centrifuged for 45 min at 20,000 × g at 1 °C. Supernatant was loaded onto a 5 ml GSTrap 4B column (GE) in a superloop for 1.5 h and the column was washed in Buffer A (20 mM Tris pH 8.0, 100 mM NaCl, 5% (v/v) glycerol, 2 mM BME) to remove non-specifically bound proteins. The GST tag was cleaved by adding Buffer A containing 10 mM BME and TEV protease to the column and incubating overnight at 4 °C. Cleaved protein was eluted with Buffer A. Protein was further purified by separating on a 5 ml HiTrap Q column with a gradient of Buffer A and Buffer B (20 mM Tris pH 8.0, 1 M NaCl, 5% (v/v) glycerol, 2 mM BME). Protein was pooled and concentrated using an Amicon 30 K concentrator, and was flash-frozen in liquid nitrogen and stored at −80 °C. A SDS-PAGE gel of each purified Rab GTPase is shown in Supplemental Fig. 1.

**In vitro GEF assay.** C-terminally His-tagged Rab was purified as described above. Each Rab was preloaded for the assay by adding EDTA to a final concentration of 5 mM and incubating for 30 min prior to adding 5-fold excess of Mant-GDP (ThermoFisher Scientific). Magnesium chloride was added to 10 mM to terminate the loading process and the solution was incubated for 30 min at 25 °C. Size exclusion chromatography was performed using a Superdex 75 10/300 column in SEC Buffer 2 (20 mM HEPES pH 7.5, 150 mM NaCl, 1 mM MgCl$_2$, 0.5 mM TCEP) to remove any unbound nucleotide. Fractions containing Mant-GDP loaded Rab were pooled, concentrated, flash-frozen in liquid nitrogen, and stored at −80 °C. Reactions were conducted in 10 µl volumes with a final concentration of 4 µM Mant-GDP loaded Rab, 100 µM GTPγS, and TRAPPII (19–300 nM). Rab and membrane (0.2 mg/ml) were aliquoted into a 384-well, black, low-volume plate (Corning 3676). To start the reaction, TRAPPII and GTPγS were added simultaneously to the wells and a SpectraMax® M5 Multi-Mode Microplate Reader was used to measure the fluorescent signal for 1 h (excitation λ = 366 nm; emission λ = 443 nm). The reaction was carried out at 25 °C. Data were analyzed using GraphPad Prism 7 Software, and $k_{cat}/K_m$ analysis was carried out according to the protocol of Delprato et al.[69]. Data were collected and exported using Softmax Pro 6.2.1. GEF curves were fit to a non-linear dissociate one-phase exponential decay using the formula $I(t) = (I_0 - I_\infty)*\exp(-k_{obs}*) + I_\infty$ (GraphPad Software), where $I(t)$ is the emission intensity as a function of time, and $I_0$ and $I_\infty$ are the emission intensities at $t = 0$ and $t = \infty$. The catalytic efficiency $k_{cat}/K_m$ was obtained by a slope of a linear least-squares fit to $k_{obs} = k_{cat}/K_m*[GEF] + k_{intr}$, where $k_{intr}$ is the rate constant in the absence of GEF.

**Lipid vesicle preparation.** Two different nickelated lipid vesicles (NiV) were prepared. No charge NiV were made with [97.5% phosphatidylcholine (egg yolk PC Sigma) and 2.5% DGS NTA(Ni) (18:1 DGS NTA(Ni), Avanti), while charged NiV were made with [20% phosphatidylserine (bovine brain PS, Sigma), 10% L-α-phosphatidylinositol-4-phosphate (PI4P, Avanti) 67.5% phosphatidylcholine (egg yolk PC Sigma), and 2.5% DGS NTA(Ni) (18:1 DGS NTA(Ni), Avanti)]. Vesicles were prepared by combining liquid chloroform stocks together at appropriate concentrations and evaporating away the chloroform with nitrogen gas. The resulting lipid film layer was desiccated for 20 min before being resuspended in lipid buffer (20 mM HEPES (pH 7.5) and 100 mM KCl) to a concentration of

1 mg/ml. The lipid solution was vortexed for 5 min, bath sonicated for 10 min, and flash-frozen in liquid nitrogen. Vesicles were then subjected to three freeze-thaw cycles using a warm water bath. Vesicles were extruded 11 times through a 400-nm NanoSizer Liposome Extruder (T&T Scientific) and stored at −80 °C.

**Mapping of the TRAPPII-Rab-binding interfaces using HDX-MS**. HDX reactions were conducted in 20 µl reaction volumes with a final concentration of 1 µM Rab1, Rab11, or Rab43 and 350 nM TRAPPII per sample. Exchange was carried out in triplicate for four time points (3, 30, 300, and 3000 s at room temperature). Prior to the addition of $D_2O$, proteins were incubated on ice in the presence of 20 mM EDTA for 30 min to facilitate release of nucleotide. Hydrogen deuterium exchange was initiated by the addition of 17 µl of $D_2O$ buffer solution (10 mM HEPES pH 7.5, 50 mM NaCl, 97% $D_2O$) to 3 µl of the protein solutions, to give a final concentration of 85% $D_2O$. Exchange was terminated by the addition of acidic quench buffer at a final concentration 0.6 M guanidine-HCl and 0.9% formic acid. Samples were immediately frozen in liquid nitrogen at −80 °C.

**HDX-MS data analysis**. Protein samples were rapidly thawed and injected onto an ultra-performance liquid chromatography (UPLC) system kept in a cold box at 2 °C. The protein was run over two immobilized pepsin columns (Applied Biosystems; Porosyme 2-3131-00) and the peptides were collected onto a VanGuard Precolumn trap (Waters). The trap was eluted in line with an ACQUITY 1.7 µm particle, $100 \times 1 \; mm^2$ C18 UPLC column (Waters), using a gradient of 5–36% B (Buffer A 0.1% formic acid, Buffer B 100% acetonitrile) over 16 min. Mass spectrometry experiments were performed on an Impact QTOF (Bruker), and peptide identification was done by running tandem mass spectrometry (MS/MS) experiments run in data-dependent acquisition mode. The resulting MS/MS datasets were analyzed using PEAKS7 (PEAKS), Peptides were identified using a target decoy database search in the software program PEAKS7. The database was composed of common affinity purification protein contaminants, other proteins purified in the lab, pepsin, and TRAPP subunits. The search parameters were set with a precursor tolerance of 20 ppm, fragment mass error 0.02 Da, charge states from 1 to 8, a false discovery rate of 0.9%, leading to a selection criterion of peptides that had a $-10 \, logP$ score of 24.3. HDExaminer Software (Sierra Analytics) was used to automatically calculate the level of deuterium incorporation into each peptide. All peptides were manually inspected for correct charge state and presence of overlapping peptides. Deuteration levels were calculated using the centroid of the experimental isotope clusters. Differences in exchange in a peptide were considered significant if they met all three of the following criteria: >4% change in exchange, >0.4 Da difference in exchange, and a $p$ value < 0.01 using a two-tailed student $t$-test. The raw HDX data are shown in two different formats. The raw peptide deuterium incorporation graphs for a selection of peptides with significant differences are shown in Fig. 3e, with the raw data for all analyzed peptides in the source data. To allow for visualization of differences across all peptides, we utilized number of deuteron difference (#D) plots (Fig. 3f + Supp. Fig. 2). These plots show the total difference in deuterium incorporation over the entire H/D exchange time course, with each point indicating a single peptide.

Samples were only compared within a single experiment and were never compared to experiments completed at a different time with a different final $D_2O$ level. The data analysis statistics for all HDX-MS experiments are in Supplemental Table 2 according to the guidelines of Masson et al.[70]. The mass spectrometry proteomics data have been deposited to the ProteomeXchange Consortium via the PRIDE partner repository[71] with the dataset identifier PXD020890.

**Negative-stain EM and image analysis of TRAPPII**. Purified TRAPPII was adsorbed to glow discharged copper grids coated with continuous carbon then stained with uranyl formate. The stained specimens were examined using a Talos L120C transmission electron microscope (ThermoFisher Scientific) operated at an accelerating voltage of 120 kV and equipped with a Ceta charged-coupled-device (CCD) camera. Two hundred and fifty micrographs were acquired at a nominal magnification of ×45,000 at a defocus of approximately −1 µm and binned twice to obtain a final pixel size of 4.53 Å/pixel. Contrast-transfer function (CTF) parameters of each micrograph were estimated using CTFFIND4[72]. Two hundred particles were manually picked then aligned to generate 2D class averages in Relion 3.0[73]. These averages were used as a template to autopick 72,731 particles, which were then aligned and classified to calculate 2D class averages. Selection of the best 2D classes yielded a particle count of 27,510. These particles were then imported to cryoSPARCv2[74] and used for ab initio reconstruction of three 3D models. The class representing the complete TRAPPII assembly contained 11,959 particles and was subsequently subjected to homogenous refinement, yielding a final reconstruction at a resolution of 19.5 Å, as calculated at 0.143 criterion using the gold-standard method. A model of the mammalian TRAPP core was generated using crystal structures of yeast TRAPPI (3CUE), human TRAPPC6a (2J3T), and mouse TRAPPC2 (2J3W) and fitted into the EM density map using UCSF Chimera[75].

For the GST-Rab43-TRAPPII complex, negative-stain samples were prepared and data were collected as described above. Four hundred and seventy particles were manually picked from 70 micrographs and subjected to 2D classification using Relion 3.0. Two 2D classes containing 362 particles showed clear density corresponding to GST-Rab43 which is not seen in the apo-TRAPPII sample.

A Gold-standard Fourier shell correlation curve showing the resolution of the TRAPPII model is shown in Supplemental Fig. 4.

**Chemical cross-linking and digestion**. Cross-linking was performed with either succinimidyl 4,4′-azipentanoate (SDA, Thermo Scientific) or succinimidyl 6-(4,4′-azipentanamido)hexanoate (LC-SDA, Thermo Scientific) reagent at an equimolar ratio with total TRAPP lysines in a buffer consisting of 20 mM HEPES pH 7.5, 150 mM NaCl, 4.6% glycerol, 0.02% CHAPS, and 0.5 mM TCEP, following the method described in ref. [57]. Briefly, available lysines were coupled to the reagent in a 10 min reaction at room temperature followed by 5 s of pulsed laser photolysis at 355 nm. Reactions were then quenched with 50 mM ammonium bicarbonate. Cross-linked samples were denatured, alkylated with 40 mM chloroacetamide (Sigma), and then digested with trypsin (MS-grade; Thermo Scientific) at a 1:20 protein-to-enzyme ratio for 4 h at 37 °C. Digestion was quenched with 0.5% formic acid (Thermo Scientific) and size exclusion chromatography (SEC) used to enrich for cross-linked peptides (Superdex Peptide PC 3.2/30 SEC column, GE Healthcare), using a mobile phase of 30% ACN, 0.1% formic acid. Fractions were collected, lyophilized, and resuspended in 0.1% formic acid for LC–MS/MS analysis.

**LC–MS/MS data acquisition and analysis**. Samples were injected on a nLC-1200 (Thermo Scientific) equipped with an Acclaim PepMap 100 guard column (75 µm × 2 cm C18, 3 µm particles, 100 Å; Thermo Scientific) and separated on a 50 cm PepMap RSLC C18 column (75 µm × 50 cm, 2 µm particles, 100 Å; Thermo Scientific). Peptides were eluted using a 60 min gradient (10–65%B) at 300 nL/min. Mobile phase A consisted of 0.1% v/v formic acid in 3% ACN, mobile phase B consisted of 0.1% v/v formic acid in 80% ACN. Data were acquired with an Orbitrap Fusion Lumos (Thermo Scientific) in OT/OT mode. Spray voltage was set at 2.0 kV with a transfer capillary temperature of 300 °C. MS was acquired with a resolution of 120,000 and mass range of 350–1250 Th. The most intense ions with charge states 4–8 within a 3-s cycle were selected for fragmentation via HCD (NCE of 32 and isolation width of 1.5 $m/z$ and a 30-s dynamic exclusion). MS/MS data were acquired at a resolution of 15,000 with 100 ms maximum injection time and a target AGC of $1.0 \times 10^5$. All data were analyzed using the cross-linking module (CRIMP) in MS Studio v2.3.0 (www.msstudio.ca)[76]. Parameters were set as follows: charge states 4–8, peptide length 4–60, percent $E$-value threshold = 50, MS mass tolerance = 5 ppm, MS/MS mass tolerance = 10, elution width = 0.25 min. Cross-linked residue pairs were constrained to K on one end and one of any amino acid on the other. Identifications were truncated with a 0.1% FDA and remaining identifications were manually validated. XL data were exported with redundancy and ambiguity reduction filters (RT grouping = 0.75 min, delta score = 12, gamma score = 0.0). Cross-links were visualized with the aid of the XlinkAnalyzer plugin in Chimera[77]. All peptides are shown in Supplemental Data 2.

**Statistics and reproducibility**. For GEF and HDX-MS assays, experiments were carried out in triplicate and means ± SD are shown in figures. Statistical analysis between conditions was performed using a two-tailed Student's $t$-test.

**Reporting summary**. Further information on research design is available in the Nature Research Reporting Summary linked to this article.

## Data availability

The mass spectrometry proteomics data have been deposited to the ProteomeXchange Consortium via the PRIDE partner repository[71] with the dataset identifier PXD020890. The processed HDX-MS data are provided as Supplementary Data 1. All data used to generate main text figures are shown in Supplementary Data 2. All other data are available from the corresponding author on reasonable request.

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

## Acknowledgements

J.E.B. is supported by a new investigator grant from the Canadian Institute of Health Research (CIHR), a discovery research grant and an accelerator supplement from the Natural Sciences and Engineering Research Council of Canada (NSERC-2014-05218 and NSERC-2020-04241), and a Michael Smith Foundation for Health Research (MSFHR) Scholar award (17686). M.L.J. is supported by a NSERC CGSM. C.K.Y. is supported by Foundation Grant from CIHR (FDN-143228). U.D. is supported by a NSERC PGSD. D.J.S. is supported by NSERC 2017-04879.

## Author contributions

J.E.B. and M.L.J. designed all biophysical/biochemical experiments. M.L.J., N.J.H., and E.M.M. carried out protein expression/purification. M.L.J, N.J.H, and E.M.M. carried out all biochemical studies. U.D. and C.K.Y. performed negative-stain EM experiments and data analysis. D.S.Z. and D.C.S. carried out chemical cross-linking experiments. A.R. and D.C.S. carried out chemical cross-linking analysis. K.D.F. and M.L.J. carried out HDX-MS experiments. J.E.B., M.L.J, and N.J.H. wrote the manuscript, with input from all authors.

## Competing interests
The authors declare no competing interests.
