## [Peer Review File · Communications Biology]

Reviewers' comments:

Reviewer #1 (Remarks to the Author):

This is a timely and interesting analysis of the specificity of human TRAPPII coupled to HDX-MS, crosslinking MS, and low resolution EM structural analysis, which complements and extends pre-existing studies of the yeast TRAPP complexes. It is recommended for publication following attention to these relatively minor points:

The authors indicated that >4% is the significant decrease in deuterium incorporation, however in Figure 3F and supplementary Figure 1A, this criteria only fits the data of TRAPPC4 in presence of Rab43 but not the remaining data which are < 2%. This is confusing.

In Figure 3C, the protection and deprotection regions are mapped onto the structure of TRAPPC2L, however these changes are not observed in the difference plots Figure 3F and Supplementary Figure 1A. The cross-linking data demonstrated that TRAPPC2L interacts with TRAPPC6A and TRAPPC3, which are not in proximity with canonical binding site, but there is detected protection and deprotection on this subunit, why?

Please provide SDS-PAGE gels of all the tested small GTPases.

In Figure 3B, please provide an additional lane showing GST-Rab43 for comparison.

The protection detected on the TRAPP5 subunit is possibly induced by the movement of the C terminus of TRAPP4 (res 181-191) when it stimulates the nucleotide release of small GTPase, but is not necessarily part of the binding site with small GTPases.

Typo, line 564, 19 nM or 9 nM?

In supplementary Fig. 2B, the blue crosslinks overlap with TRAPPC3 structure and it is difficult to read. Please change the crosslinks to another color.

Reviewer #2 (Remarks to the Author):

The manuscript by Jenkins et al reports a biochemical and preliminary structural characterization of the mammalian TRAPPII complex. While the yeast and drosophila TRAPPII complexes have been investigated in more detail before, information on the mammalian complex is limited. The authors establish the purification of human TRAPPII expressed in insects cell and use this preparation for in vitro GEF assays. They confirm that TRAPPII is a GEF for Rab1, Rab 11 (isoforms a, b, but not c) and newly identify Rab19 and Rab43 as substrates. HDX-MS experiments suggest that all GTPases bind to the same active site, which is also confirmed by structural comparison of apo-TRAPPII and the Rab43-TRAPPII complex by low resolution negative stain EM. A 3D reconstruction of TRAPPII combined with chemical crosslinking data provides a rough model of mammalian TRAPPII architecture. Finally, several variants of TRAPPII substrate Rabs are investigated.

This study provide interesting new insight into the biochemical properties of TRAPPII and the cognate Rab GTPases. Several minor points should be addressed prior to publication:

1. It would be interesting to compare the catalytic efficiency of TRAPPII for Rab19 with the results for Rab 1, 11 and 43. I recommend to include the calculation of k_{cat}/K_M in fig 2d.
2. Why is Rab11c/35 not a substrate? Can the authors use the model of TRAPP/Rab11 (fig 5a) to provide a molecular explanation.
3. It would also be more appropriate to compare the structure of SH3BP/Rab11 to the model of TRAPP/Rab11, rather than to a TRAPP/Rab1 model in fig 5f.
4. Is there an explanation why differences in HDX were observed for C2L, but not C1? The model

of TRAPP_{II} subunit architecture (fig 1c) appears inconsistent with the HDX results.
5. The FSC plot should be included in the supplement

Reviewer #3 (Remarks to the Author):

The manuscript by Jenkins et al. reports a number of interesting observations about the biochemical properties and structural organization of the purified mammalian TRAPP2 complex. GEF assays for a panel of mammalian Rab GTPases identify Rabs 1, 11, 19 and 43 as in vitro substrates in solution and on membranes. GEF activity for Rab43 and the closely related Rab19 has not been previously described. HDX-MS experiments indicate that the GEF site is associated with the TRAPP core rather than TRAPP_{II}-specific subunits and coincides with that for Rab1 and Rab11. This result is supported by NS-EM and XL-MS experiments, which further indicate a structural organization similar to TRAPP_{II} from flies and yeast, despite being monomeric rather than dimeric. Finally, the authors characterize the effects of a 'TRAPPopathy'-like Rab11 mutation on GEF activity and engineer a structured-based Rab11 mutation that reduces GEF activity with respect to SH3BP5 while maintaining TRAPP_{II} GEF activity. Together these and other results provide important new insights into mammalian TRAPP_{II} that will be of interest to the field.

Overall, the manuscript is well written and the experiments appear to be well executed and appropriately analyzed. I have only a few relatively minor comments.

Do the authors think the increased GEF activity in the presence of membranes containing anionic phospholipids is a consequence of increased membrane recruitment of TRAPP_{II} or is there evidence that would suggest a different mechanism?

The Rab11 K58 mutation is potentially interesting as a tool for delineating the role of activation by TRAPP_{II} vs. SH3BP5, assuming the mutation does not interfere with effector binding, GAP activity or interactions with other binding partners (e.g. RabGDI, PI4KIII_B) Have the authors considered whether the mutation would be likely or not to alter interactions with effectors or other binding partners based on available structural information? Might be worth discussing.

Line 140, "consistent with distinct from" appears to be a typo. Probably "distinct from" was meant.

Summary of reviewer reports:

Referee #1: Recommends for publication following few clarifications and explanations as well as providing SDS PAGEs of the studied proteins.

We have added all clarifications and explanations (see details below), with SDS-page gels of all studied proteins now added

Referee #2: Requests to address a few minor points including k_{cat}/K_M calculations and explanations for several the HDX results.

We have added all minor points (see details below), along with describing the k_{cat}/K_M calculations for Rab19

Referee #3: Requests explanations for the recruitment of TRAPP^{II} to membranes and to further discuss the interaction properties of the Rab11 K58 mutation based on the provided structural data.

We have added description of the membrane recruitment and the details on how the Rab11 K58 mutant would affect other molecular interactions (see details below and new panel in Fig 5).

Reviewers' comments:

Reviewer #1 (Remarks to the Author):

This is a timely and interesting analysis of the specificity of human TRAPP^{II} coupled to HDX-MS, crosslinking MS, and low resolution EM structural analysis, which complements and extends pre-existing studies of the yeast TRAPP complexes. It is recommended for publication following attention to these relatively minor points:

We appreciate the positive comments from the reviewer.

The authors indicated that >4% is the significant decrease in deuterium incorporation, however in Figure 3F and supplementary Figure 1A, this criteria only fits the data of TRAPPC4 in presence of Rab43 but not the remaining data which are < 2%. This is confusing.

The criteria for a change in deuterium incorporation to be considered significant is that the peptide has to meet the following three criteria: >4%, >0.4Da, and a two tailed T-test p value <0.01. The reason we use a criteria that uses both percentage and number of deuterons is that either of these metrics bias towards short or long peptides, respectively. An example of the raw data is shown in Fig 3E. These graphs are the most informative but it is impossible to show this for all ~450 peptides analysed.

The graph shown in Fig 3F and Supp Fig 1A are #D difference graphs. In this graph every point represents an analyzed peptide, with the y axis being the sum of the #D difference across all time points. The change seen here is actually a #D difference of over 4 Da with Rab43 occurring at the N-terminus of TRAPPC4 (see peptide 5-19 in panel 1E), which corresponds to a %D difference of >15% (see Fig. 1D). We agree that these graphs can

be less informative than the individual peptide graphs, but it allows us to show the entire dataset (and the error associated) in a single main text figure.

We have added additional description in the methods to clarify this point.

In Figure 3C, the protection and deprotection regions are mapped onto the structure of TRAPPC2L, however these changes are not observed in the difference plots Figure 3F and Supplementary Figure 1A. The cross-linking data demonstrated that TRAPPC2L interacts with TRAPPC6A and TRAPPC3, which are not in proximity with canonical binding site, but there is detected protection and deprotection on this subunit, why?

I think this concern is mainly addressed by the comments above. TRAPPC2L has significant differences (80-88, 100-110).

We agree that it is intriguing that there are conformational changes that occur in TRAPPC2L, as this is expected to be quite distant from the Rab binding site. An important note is that the changes in TRAPPC2L are much smaller than the corresponding changes in the Rab binding site (see peptides in Fig 3E).

These changes are likely due to an allosteric effect, however, with our current low resolution structural data we can not clearly define the molecular mechanism of this change.

We have added in description of this in the results.

Please provide SDS-PAGE gels of all the tested small GTPases.

All SDS-page gels of all studied proteins are now included in the supplement (See new Supplemental Fig 1).

In Figure 3B, please provide an additional lane showing GST-Rab43 for comparison.

We have added in an additional SDS-page gel showing the purity of the GST-Rab43 construct (Supplemental Fig 1).

The protection detected on the TRAPP5 subunit is possibly induced by the movement of the C terminus of TRAPP4 (res 181-191) when it stimulates the nucleotide release of small GTPase, but is not necessarily part of the binding site with small GTPases.

This is an excellent point, the region protected in TRAPPC5 would not be predicted to be in contact with Rab11 or Rab43, but is in contact with the region 181-195 of TRAPPC4 that directly participates in stabilising the Nucleotide free Rab. We agree that this likely an allosteric effect due to the altered conformation in TRAPPC4, leading to a change in TRAPPC5.

We have included discussion of this point in the results.

Typo, line 564, 19 nM or 9 nM?

This is a typo, it is meant to be 19 nM, this has been changed in text.

In supplementary Fig. 2B, the blue crosslinks overlap with TRAPPC3 structure and it is difficult to read. Please change the crosslinks to another color.

We agree that the crosslinks shown in Supp Fig 2 are hard to read in the current color scheme. The crosslinks are in a fixed color due to the program used, but we have changed the color of the protein subunits, so the crosslinks are very clear against the new background. See new Supp Fig. 3.

Reviewer #2 (Remarks to the Author):

The manuscript by Jenkins et al reports a biochemical and preliminary structural characterization of the mammalian TRAPPII complex. While the yeast and drosophila TRAPPII complexes have been investigated in more detail before, information on the mammalian complex is limited. The authors establish the purification of human TRAPPII expressed in insects cell and use this preparation for in vitro GEF assays. They confirm that TRAPPII is a GEF for Rab1, Rab 11 (isoforms a, b, but not c) and newly identify Rab19 and Rab43 as substrates. HDX-MS experiments suggest that all GTPases bind to the same active site, which is also confirmed by structural comparison of apo-TRAPPII and the Rab43-TRAPPII complex by low resolution negative stain EM. A 3D reconstruction of TRAPPII combined with chemical crosslinking data provides a rough model of mammalian TRAPPII architecture. Finally, several variants of TRAPPII substrate Rabs are investigated.

This study provide interesting new insight into the biochemical properties of TRAPPII and the cognate Rab GTPases. Several minor points should be addressed prior to publication:

We appreciate the positive comment from the reviewer.

1. It would be interesting to compare the catalytic efficiency of TRAPPII for Rab19 with the results for Rab 1, 11 and 43. I recommend to include the calculation of k_{cat}/K_M in fig 2d. This is an excellent suggestion from the reviewer. We attempted to measure the k_{cat}/K_M values of Rab19 using varying concentrations of TRAPPII. Unexpectedly we found a non-linear response in the observed GEF rate upon varying TRAPP concentrations (See Fig 2 Panel B). This makes a calculation of the k_{cat}/K_M values impossible against Rab19. This suggests that there may be a unknown concentration driven effect that will require further study to dissect.

We have included a discussion of this in the results.

2. Why is Rab11c/35 not a substrate? Can the authors use the model of TRAPP/Rab11 (fig 5a) to provide a molecular explanation.

It was unexpected that there was no activity against Rab25 while there was robust activity against Rab11. Our initial cursory analysis of the yeast structure of Rab1(ypt1) bound to TRAPPII, and the alignment of the mammalian variants of Rab25 with Rab11/Rab1 did not provide any clear insight.

Upon the reviewers suggestion we have done a more careful analysis of the sequence difference between Rab11A and Rab25. There are three main regions of Rab11 predicted to interact with the TRAPP core, including the N-terminal beta strand, and the two switch regions. Almost all of the contact residues in the switches are conserved between Rab11 and Rab25. Upon closer inspection the most divergent region between Rab25 and the substrates Rab (Rab1, Rab11, Rab19, and Rab43) is the n-terminus.

Further experimentation will be needed to verify this as the mechanism of specificity, and are excellent ideas for further studies, however, we feel this is out of the scope of the current manuscript.

We have added text describing this in the discussion section.

3. It would also be more appropriate to compare the structure of SH3BP/Rab11 to the model of TRAPP/Rab11, rather than to a TRAPP/Rab1 model in fig 5f.

This is a good suggestion, however, the main problem with this is that there is no high or medium resolution structure of TRAPP_{II} bound to Rab11. As there is significant conformational rearrangement of many of the regions in contact, in particular lysine 58 in Rab11 (55 in Ypt1) which is pointed towards solvent in the free Rab11 structures that could be used as a model.

This would give a false impression of the atomic position of this residue when bound to TRAPP_{II}. We agree that an optimal situation would be to use a structure of Rab11 bound to TRAPP, however, this does not currently exist.

4. Is there an explanation why differences in HDX were observed for C2L, but not C1? The model of TRAPP_{II} subunit architecture (fig 1c) appears inconsistent with the HDX results.

It was unexpected to see a conformational change in TRAPP_{C2L} when in complex with Rab43. This is likely an allosteric effect, for which we can't currently propose a convincing molecular mechanism. An important note is that the changes in TRAPP_{C2L} are much smaller than the corresponding changes in the Rab binding site (see peptides in Fig 3E).

The reviewer is correct that it might be expected to see a conformational change in TRAPP_{C1}, as the helix from 32-52 in TRAPP_{C1} (specifically residues 40,41 and 43) would be expected to be in contact with switch I of the bound Rab. From examining the raw HDX exchange data (source data) a peptide covering this helix (35-42) is extremely stable with deuterium levels only reaching ~10% at 3000 s of exchange. There is protection with Rabs bound (~4% with all tested Rabs). This meets the t-test for significance, but does not cross the #D threshold. This can be a shortcoming of HDX experiments is that it is very difficult to map protein binding sites in extremely stable regions of a protein.

We have included additional discussion of this in the results.

5. The FSC plot should be included in the supplement

This has been added (see Supp. Fig 4).

Reviewer #3 (Remarks to the Author):

The manuscript by Jenkins et al. reports a number of interesting observations about the biochemical properties and structural organization of the purified mammalian TRAPP2 complex. GEF assays for a panel of mammalian Rab GTPases identify Rabs 1, 11, 19 and 43 as in vitro substrates in solution and on membranes. GEF activity for Rab43 and the closely related Rab19 has not been previously described. HDX-MS experiments indicate that the GEF site is associated with the TRAPP core rather than TRAPP-II-specific subunits and coincides with that for Rab1 and Rab11. This result is supported by NS-EM and XL-MS experiments, which further indicate a structural organization similar to TRAPP-II from flies and yeast, despite being monomeric rather than dimeric. Finally, the authors characterize the effects of a 'TRAPPopathy'-like Rab11 mutation on GEF activity and engineer a structured-based Rab11 mutation that reduces GEF activity with respect to SH3BP5 while maintaining TRAPP-II GEF activity. Together these and other results provide important new insights into mammalian TRAPP-II that will be of interest to the field.

Overall, the manuscript is well written and the experiments appear to be well executed and appropriately analyzed. I have only a few relatively minor comments.

We appreciate the positive comments from the reviewer.

Do the authors think the increased GEF activity in the presence of membranes containing anionic phospholipids is a consequence of increased membrane recruitment of TRAPP-II or is there evidence that would suggest a different mechanism?

At this stage we have no additional evidence that the increased activity is distinct from increased membrane binding.

Evidence from Thomas et al JCB 2016 are quite convincing that TRAPP-II membrane recruitment is strongly driven by anionic lipids, with this proposed to be a primary factor in Rab activation. We agree with this hypothesis.

We are planning to initiate HDX-MS experiments examining possible conformational changes that occur upon membrane binding, that may propagate to the Rab binding site. These experiments hopefully will lead to insight on this process, but are outside of the scope of the current manuscript.

We have added additional text in the discussion on this point.

The Rab11 K58 mutation is potentially interesting as a tool for delineating the role of activation by TRAPP-II vs. SH3BP5, assuming the mutation does not interfere with effector binding, GAP activity or interactions with other binding partners (e.g. RabGDI, PI4KIIIIB) Have the authors considered whether the mutation would be likely or not to alter interactions with effectors or other binding partners based on available structural information? Might be worth discussing.

This was an excellent suggestion from the reviewer. We have added in a new panel locating K58 on the structures/structural models of the following binding partners: Rab-

GAPs, Rab-Escort, and Rab effectors/binders (PI4KB, Rabin8, FIP3). This clearly shows that this residue is distant from GAP, escort, and effector binding sites. This highlights that this is likely a residue that would have limited effects on other Rab11 functions other than modulating selectivity for TRAPP11 versus SH3BP5.

We have added in additional discussion of this in the results.

Line 140, "consistent with distinct from" appears to be a typo. Probably "distinct from" was meant.

This was indeed a typo and has been fixed in the new manuscript. The intended meaning was that the